# A Comprehensive Review on Photoacoustic-Based Devices for Biomedical Applications

**DOI:** 10.3390/s22239541

**Published:** 2022-12-06

**Authors:** Rita Clarisse Silva Barbosa, Paulo M. Mendes

**Affiliations:** CMEMS-UMinho, University of Minho, 4710-057 Braga, Portugal

**Keywords:** microfabrication, biomedical probes, multiphysics modeling, miniaturized ultrasound probes, photoacoustic effect, photoacoustic imaging

## Abstract

The photoacoustic effect is an emerging technology that has sparked significant interest in the research field since an acoustic wave can be produced simply by the incidence of light on a material or tissue. This phenomenon has been extensively investigated, not only to perform photoacoustic imaging but also to develop highly miniaturized ultrasound probes that can provide biologically meaningful information. Therefore, this review aims to outline the materials and their fabrication process that can be employed as photoacoustic targets, both biological and non-biological, and report the main components’ features to achieve a certain performance. When designing a device, it is of utmost importance to model it at an early stage for a deeper understanding and to ease the optimization process. As such, throughout this article, the different methods already implemented to model the photoacoustic effect are introduced, as well as the advantages and drawbacks inherent in each approach. However, some remaining challenges are still faced when developing such a system regarding its fabrication, modeling, and characterization, which are also discussed.

## 1. Introduction

The photoacoustic effect consists of the process by which acoustic waves are generated as a result of light incidence on a material with specific properties. The incidence of light on the material raises its temperature, and as a result, it thermally expands. The continuous thermal expansion and retraction will then generate ultrasound (US) waves.

This phenomenon has been widely explored for two main purposes: to construct a highly miniaturized US transmitter, which includes the photoacoustic target in the probe itself [1], and to perform photoacoustic imaging (PAI), where the photoacoustic target is the tissue that is being examined [2]. The former application can be extremely valuable to conduct minimally invasive surgeries, which have multiple advantages, such as reduced blood loss, surgical trauma, and risk of postoperative complications, as well as improved recovery times and cosmesis results, and they can be either highly beneficial for obese people or patients with high-risk factors to undergo open surgery [3,4]. The latter avails the important information provided by optical methods since light is transmitted, which is merged with the high spatial resolution, deep penetration, and the low scattering, refraction, and attenuation of the acoustic waves. Moreover, the received US waves allow us to perceive the positioning of the studied structures while granting the contrast and depth of the image, unlike the pure optical imaging methods [5,6,7,8,9,10,11,12]. In addition, there is another potential application that consists of using high intensity and focused US for targeted therapy [13]. These advantages render the photoacoustic effect promising for sundry biomedical applications.

A wealth of devices has already been developed using this technology in research settings, both to perform PAI and to develop US probes. PAI provides important clinical information that can have great diagnostic relevance regarding various diseases in different organs, such as ischemia [6,14], tumors [12,15,16], neurodegenerative diseases [6], epilepsy [6,12], and stroke [6,17] through brain imaging. This imaging modality can also serve as an angiography technique, thus enabling vasculature imaging and determination of blood oxygenation due to distinct acoustic signals produced by oxyhemoglobin and deoxyhemoglobin [18,19,20,21,22]. Besides the detection of vascular diseases through vascular imaging, this imaging technique could also be useful for tracking hemoglobin uptake, spotting regions with rapidly growing cells that are cancer indicators, and assessing tumor response to therapies [23,24,25,26,27,28,29]. Tumor metastases can also be ascertained by sentinel lymph node imaging [12,30]. Breast imaging is another widely explored PAI modality, and it is already under clinical studies, particularly for cancer detection [11,31,32,33,34,35]. This emerging imaging technique has the potential for high-resolution dermatologic imaging, namely, to diagnose skin melanomas [26,36,37], carcinomas [11], psoriasis [11,12,36], atopic dermatitis [36,37], burn injuries [12,36], and bacterial wound infection [11,36]. Thyroid imaging [12,38,39,40,41], reproductive [12,21,42,43] and urological [7,44] systems imaging, neonatal imaging [7,45,46], gastrointestinal imaging [44,47,48,49], adipose tissue imaging [41,50], musculoskeletal imaging [44,51], ophthalmologic imaging [52], and even diagnosis of COVID-19 [53] have also been successfully performed by exploring the photoacoustic effect. Furthermore, PAI can also be useful for biopsy guidance [43,49], image-guided therapy [54], drug delivery [54,55], and intraoperative imaging [55,56,57]. This imaging technique’s potential can be further enhanced when combined with other imaging modalities such as US [41,57,58], optical coherence tomography [49], fluorescence [49,58], and electromagnetic sensing [53]. M-Mode [59], 2D [60,61,62], 3D [61,63,64], and B-Mode images [1,65] of biological tissues have already been acquired, as shown in Figure 1, demonstrating the feasibility of the photoacoustic effect to develop US probes. One key application of the photoacoustic effect is in the biomedical field; notwithstanding, it is also suitable for other nondestructive testing and evaluation [62], and for industrial applications [66].

The implementation of a system relying on the photoacoustic effect is possible on materials exhibiting photoacoustic properties and requires the use of a light source with appropriate properties to optimize performance for the material under test and an acoustic detector with appropriate bandwidth. When a US probe based on the photoacoustic effect is being developed, it is also important to include a strategy for steering the US beam in the system. The probe is required when either the material under observation does not exhibit a photoacoustic effect or we want to control some system feature, such as frequency of operation or US power, or when we need to control spatial and time resolution of obtained images.

Photoacoustics is a highly disciplinary research field, from materials to optics and acoustics. In this way, the full know-how to tackle the development of such devices is widespread among a wide variety of papers. In this paper, such information was collected and placed together to be much more helpful as an advanced starting point for anyone intending to implement such a system.

The great relevance of the photoacoustic effect, mainly for biomedical applications, is highlighted throughout this review. In addition, the diversity of materials that can act as photoacoustic targets and the components that have already been used to devise a system relying on the photoacoustic effect are also presented. By reviewing the various photoacoustic-based devices that have already been developed, a comparative analysis can be conducted between the different systems. This can be an asset to conclude which components should be included as providing better performance to perform PAI or to build a highly miniaturized ultrasound probe. For a deeper insight when developing the system, the photoacoustic effect should be modeled. This can be accomplished either by mathematical modeling or by using simulation software, following the essential steps detailed later. Lastly, the main challenges still faced when developing such a system are outlined and discussed.

## 2. The Photoacoustic Effect and Its Potential

The photoacoustic effect was reported for the first time in 1880 by Alexander Graham Bell, the scientist known for inventing the telephone [67]. In 1878, Bell started some experiments to explore the photosensitivity of selenium in order to use this property to perform speech transmission. Finally, in 1880, Bell and his assistant Charles Sumner Tainter documented the first voice message transmission through light and wirelessly. Later this year, he exhibited a photophone, displayed in Figure 2. This system was composed of a mirror where the sunlight was focused, a movable mirror, an alum cell to prevent mirror overheating, two lenses to make the reflected beam parallel, a parabolic reflector to converge the light towards a selenium cell located in the concavity and a telephone receiver. The sound waves produced during the speech cause mirror vibrations, varying the intensity of the light detected in the receiver. In the telephone circuit, those variations in the light intensity give rise to a sound wave [12,55,67,68,69,70,71].

Bell also inadvertently noticed that when illuminating a solid with an intermittent light beam, acoustic energy is released at the modulation frequency. The photoacoustic effect was then discovered, and numerous scientists began to conduct experiments to explore this effect that allows sound encoding in the form of modulated light [55,68].

The first photoacoustic experiments in the biomedical field date back to 1964. Amar et al. sent a 400 μs pulse train with a pulse duration of 1 μs and 50 mJ per pulse to the eyes of a living rabbit onto the retinas. They reported no damage in the rabbit eyes, and the only effect claimed was eye blinking. A small US detector was located on the left occipital lobe, and it perceived acoustic waves related to the laser pulses directed to the retinas. The authors realized that these US waves were produced by light absorption in the retinas, and they would have spread through the brain to the US detector. Thenceforward photoacoustics has been showing great potential in many fields of science and medicine, and it has been widely explored for clinical roles as major technological developments are taking place. Nonetheless, the first commercially available photoacoustic-based device was only launched in 2010 [12,55,68,71,72].

Figure 3 exhibits a graphical explanation of the photoacoustic effect. At its core, the photoacoustic effect lies in the formation of a US wave from light absorption in a material with specific characteristics [73]. Thereupon, a source of modulated or pulsed light, a material/composite with both high optical absorption and high thermal expansion coefficient, and a US detector are indispensable to efficiently convert the light pulse into the US and subsequently detect it [34,50,54,63,68,74,75]. The supplied energy should be absorbed as much as possible by the target material and converted into heat, thus increasing its temperature. Hence thermoelastic expansion and retraction take place, which leads to a US wave that propagates throughout its surroundings [11,54,55,60,76,77,78].

As already noticed, one key advantage of the PAI modality is its potential as a non-destructive imaging methodology. Furthermore, since the signal generation and detection are based on light, which has an extremely small wavelength and may be transported by very thin fibers, this methodology also entails a great potential for system miniaturization. Many studies obtained photoacoustic-based transmitters with sizes between 0.84 and 2.5 mm with frequencies from 1 to 150 MHz, peak-to-peak US pressures between 2 kPa and 70 MPa, intensities of 1000–10 000 W/cm^2^, axial resolutions ranging from 40 to 380 μm, lateral resolutions from 88 to 480 μm, and penetration depth between 1 mm and 5 cm [1,59,60,63,64,75,76,79,80,81,82,83,84]. Although the developed transmitters had broad bandwidths [61,80,81], a few systems exhibited poor durability and stability as their coatings degraded with time [76].

Unlike the most conventional methods used to make US transducers (like piezo material or micromembranes), the photoacoustic effect is exclusively used for acoustic wave generation and not for detecting them. However, capacitive micromachined ultrasonic transducers (CMUTs) and piezoelectric micromachined ultrasonic transducers (PMUTs) have limited performance [60,85,86,87,88,89,90] and are typically expensive due to fabrication intricacy as size decreases related to mechanical dicing and electrical elements connecting, restricting their wide clinical applicability [60]. Therefore, photoacoustics has emerged as an option to address these drawbacks [60,83]. Furthermore, the detection of acoustic waves can also be based on optical methods, where a US wave can be converted in the variation of some optical wave feature [91].

In spite of the fact that this technology is not standardized in current imaging techniques and it still has some outstanding challenges, important research into the photoacoustic effect’s potential is still being conducted in order to demonstrate its feasibility and potential for its future common application [60,83]. Once all the challenges are addressed, the photoacoustic effect can allow the development of valuable technology for the clinical field, especially regarding minimally invasive interventions.

## 3. Acoustic Wave Generation

An efficient generation of acoustic waves urges target materials with certain properties, and the choice of a suitable photoacoustic material is a determinant to successfully generating US waves [92]. Firstly, the light energy must be greatly absorbed and converted into heat, wherefore a material with a large optical absorption coefficient is convenient [60,62,63,75]. A target material with a high thermal expansion coefficient is also mandatory to generate powerful acoustic pressures [60,62,63,75,82,83,93]. Additionally, the thickness and thermal conductivity of the material with photoacoustic properties, laser pulse width, and absorption depth are other factors that also influence photoacoustic efficiency [92].

Since the photoacoustics discovery, sundry studies have been undertaken to find out emerging materials capable of generating acoustic waves through the photoacoustic effect. This search for new materials also propelled research using biological tissues, and some authors noticed photoacoustic signals when excited by the light coming from specific molecules within the body entitled endogenous contrast agents [11,17,22,23,27,34,35,49,52,84,94,95,96]. This allows to perform PAI that provides bountiful structural, functional, molecular, and kinetic information [5,11,12,37,61,72,97]. To overcome some of the endogenous contrasts’ drawbacks, several exogenous contrasts were already explored, including various materials in different shapes, increasing the image contrast and depth [11,23,50,97]. The potential of some materials to be directly incorporated in innovative and highly miniaturized US transmitters was recognized afterward [75,83,93,98].

When aiming to develop a US probe, such as to perform PAI, the photoacoustic target must absorb the optical energy and efficiently release it in the form of acoustic energy. Although all materials that absorb optical energy eventually release acoustic energy, most of them do so inefficiently, resulting in the need to combine the optical absorber with another material with properties better suited for releasing acoustic energy.

### 3.1. Materials Explored in Photoacoustic Imaging

One key advantage of endogenous photoacoustic agents is that they are naturally inside the body, and consequently, there is no need for the introduction of foreign substances into the body. Therefore, when PAI avails these chromophores’ properties, it excludes the possibility of additional risks related to drugs [23,54].

One of the most explored endogenous contrast agents is hemoglobin, as the absorption coefficient is sensitive to its concentration and great differences are evident between the optical absorption spectrum of oxygenated and deoxygenated hemoglobin [7,12,34,54,94,95,97]. Other biological chromophores were also studied, such as water [11,54,78], melanin [11,12,95], lipids [11,34,49,50], deoxyribonucleic acid (DNA) and ribonucleic acid (RNA) [49,94], bilirubin [23,30,94], oxygenated and deoxygenated myoglobin [30,78], collagen [25,28,81], cytochromes [23,30,94], and glucose [12,30]. Figure 4 depicts the optical absorption coefficient of these biological materials as the wavelength changes. As can be seen, melanin, DNA, and RNA preferentially absorb ultraviolet radiation, while hemoglobin and bilirubin absorb in the transition zone between ultraviolet and visible radiation. Conversely, reduced myoglobin absorbs primarily visible radiation, in contrast with oxygenated myoglobin, lipids, collagen, water, and glucose, which predominantly absorb near-infrared radiation. This is relevant information in order to excite the molecules at the wavelengths where they are the more sensitive, thus obtaining stronger photoacoustic signals [71].

It is noteworthy that the applicability of these contrast agents is limited by the abundance of these substances, the lack of specificity, and the weak output intensity [23]. Exogenous contrast agents, usually injected into the bloodstream, offer advantages over endogenous ones because they can be merged with targeting molecules to selectively bind to specific receptors and provide enhanced sensitivity and contrast [23,50,54,94,99]. Some authors also demonstrated the advantages of developing contrast agents with maximum absorption in long near-infrared windows where tissue penetration is the greatest [50,54,97,99,100].

Consequently, organic dyes such as indocyanine green (Food and Drug Administration (FDA) approved [46]) and Evan’s blue [12,23,26,97], fluorescent proteins and non-fluorescent proteins [78,94,101,102], methylene blue [12,97], gold, silver, copper, tungsten, and iron oxide nanoparticles [23,50,78,97,103], carbon-based nanomaterials [27,54,97,100,104], IR800CW [27,97,105], Alexa Fluor 750 [26,106], semiconducting nanoparticles [97], and (transition metal chalcogenides)-based nanomaterials [97] have been analyzed as exogenous contrast agents. However, most of them have poor biodegradability, photostability, and clearance from the body, limiting their wide enforceability [97].

Figure 5 shows the normalized absorption spectra for some of the previously mentioned materials. The preferential absorption of infrared radiation by most exogenous contrast agents is evident in this graph, except for matrix metalloproteinase and both copper and silver nanoparticles that preferentially absorb near-infrared radiation. It is also worth mentioning that, besides the variation of the photoacoustic signal with the composition, the size, and shape of the particles also affects the output signal as well as the clearance from the body [8,23,27,50,54,97,99].

Despite the irrefutable usefulness of PAI to assess living tissues and their biological processes, most of these chromophores (excepting for carbon-based nanomaterials and metal nanoparticles) are not suitable to develop US probes due to their challenging integration, their lack of stability, and/or the complicated mimicking of the physiological conditions, especially at lower dimensions [97].

### 3.2. Optical Energy Absorption Materials for Ultrasound Probes

High ultrasound frequencies and broad bandwidths are desirable for improved axial resolutions of US probes [61,63]. It should also be emphasized that high ultrasound frequencies are greatly attenuated in the tissues resulting in low penetration depths. Thus, probes with wide bandwidths are crucial for high-resolution imaging at low depths when using high frequencies and imaging at great penetration depths when low ultrasound frequencies are applied [1,61]. Regarding the bandwidth, the optical ultrasound generation through the photoacoustic effect is an advantage over other methods since the frequencies are defined by the bandwidth of the optical excitation modulation and do not depend on resonance frequencies, so broad bandwidths can be achieved [64].

On the other hand, to reach great lateral resolutions, the ultrasound beam should be collimated or focused, which can be accomplished by employing higher frequencies and increasing the lateral dimensions of the ultrasound generator [1]. Moreover, large ultrasound pressures are relevant for ensuring the high sensitivity of the ultrasound receivers [61,63], although this is a recurrent limitation of photoacoustic-based systems [13]. Great ultrasound pressures are also crucial for the acoustic waves produced by US transmitters to reach great penetration depths [63]. This feature is typically achieved by maximizing the thickness of the target material that will generate the ultrasound wave [63,83]. However, the optimization of the ultrasound pressure needs to be balanced with the ultrasound bandwidth since large light-absorbing layers are crucial for achieving high ultrasound pressures, but it results in narrowed bandwidths due to US attenuation, decreasing the image resolution [61,62,82,109].

#### 3.2.1. Thin Films Metallic Devices

Thin films composed of metals can be used to generate acoustic waves by means of the photoacoustic effect. These thin metallic films can be built up mainly by metal evaporation [110], namely through e-beam evaporation [111].

Kozhushko and Hess used a 1 mm thick glass plate and a spherical concave stainless steel surface with a radius of 14 mm immersed in water and successfully generated acoustic waves from light [112]. Another approach to generating ultrasound through a photoacoustic effect is using thin metallic films deposited on solid substrates [93]. The potential of these films to generate ultrasound was studied in 0.15 μm thick chromium films in glass slide [82], 55–400 nm thick molybdenium film evaporated onto one face of a Pyrex wafer with optically polished faces [111], 400 nm thick aluminum film deposited onto the end surface of a sapphire rod with 4 mm and 12 mm in diameter and length respectively [113], and a 100 nm thick aluminum layer evaporated above a glass microscope slab with a thickness of 1 mm [110].

In spite of the high frequency obtained with these materials, low photoacoustic efficiency was reported. This poor performance is mostly due to the low expansion coefficient and the great reflectance of metallic films [61,93,110].

#### 3.2.2. Organic Pigments

Apart from being used to perform PAI, organic pigments can also be integrated into a probe to photoacoustically produce ultrasound. Using an optical fiber coated on the distal end with a 20 μm thick composite of crystal violet and polydimethylsiloxane (PDMS), as shown in Figure 6, a peak-to-peak pressure up to 0.90 MPa at 1.5 mm away from the transmitter and −6 dB bandwidth of 15.1 MHz were achieved with an incident optical fluence of 86.3 mJ/cm^2^, and a repetition rate of 100 Hz [61]. Nonetheless, this composite showed a loss of acoustic conversion efficiency with repeated usage due to poor photostability, limiting its usage for clinical imaging.

A plethora of other organic pigments with photoacoustic properties were already explored [9,114]. However, most of them are mainly employed to perform PAI, as already reviewed in the previous subsection.

#### 3.2.3. Metallic Nanoparticles

Fabrication of ultrasound probes based on the photoacoustic effect and including metallic nanoparticles can be carried out by a mixture of substances followed by coating [62], sputtering [61], or e-beam evaporation [109]. The selected technology depends mostly on the photoacoustic target’s requirements and the available technology.

Gold nanoparticles have demonstrated great optical absorption as well as appropriateness for biomedical applications [115]. Wu et al. used a glass slide coated with a 450 μm thick PDMS-gold nanoparticle composite and under laser energy density of 13 mJ/cm^2^, a pulse width of 150 ns, and a repetition rate of 1 kHz; they measured a 3.1 MHz bandwidth and a 189.49 kPa acoustic pressure at 1.8 mm away from the US emitter [116]. The narrowed bandwidth was associated with the long pulse width. A more recent study states that the effect of pulse duration on the frequency response of the photoacoustic signal has not yet been extensively explored, but they found a decrease in high-frequency content as pulse width increased [117]. Zou et al. coated the tip of a 400 μm diameter optical fiber with a 105 μm maximum thick composite comprising PDMS and gold nanoparticles, and the performance was significantly higher than the obtained using thin metallic films [62]. They measured a peak-to-peak pressure of 0.64 MPa at a distance of 1 mm and bandwidth over 20 MHz using a pulse duration of 5 ns, a repetition rate of 10 Hz, and a laser fluence of 8.75 mJ/cm^2^. Another study reported that an optical fiber with a 200 μm thick gold nanoparticles-PDMS coating, depicted in Figure 7, achieved a peak-to-peak pressure of 0.41 MPa at 1.5 mm and a −6 dB bandwidth of 4.5 MHz for an incident optical fluence of 55.3 mJ/cm^2^ [61]. Hou et al. also deposited by e-beam evaporation a 2 nm thick gold nanoparticle layer over a 4.5 μm thick PDMS block and detected surface pressure of 1.5 MPa and bandwidth around 65 MHz when excited by a laser with pulse energy of 100 mJ and pulse width of 5 ns [109]. The strong optical absorption of gold nanoparticles notwithstanding, the wavelength at which these particles absorb the greatest amount of light depends on their size, shape, and local dielectric environment [109,115]. Furthermore, the absorption spectrum depends on the nanoparticles’ concentration [116] and the high optical absorbing window is usually tight [61,75].

Silver, copper, tungsten, and iron oxide nanoparticles also exhibit remarkable photoacoustic properties [23,50,78,97,103]. Despite this fact, these metallic nanoparticles have been considerably more thoroughly exploited to perform PAI than to fabricate ultrasound probes relying on the photoacoustic effect.

#### 3.2.4. Carbonaceous Materials

Harnessing their strong optical absorption across the visible and near-infrared wavelength ranges, graphene [118,119], graphite [62,110], candle soot [120,121,122], carbon nanofibers [123], carbon black [64,82], and carbon nanotubes (CNTs) [1,61,124] are among the carbonaceous materials widely explored to generate acoustic waves. An optical fiber coated with a carbon film was also studied for this purpose [125]. Devices employing such materials are typically produced by microfabrication techniques, such as chemical vapor deposition [13], electrospinning [126], etching [127], and inkjet printing [63].

Reduced graphene oxide has demonstrated an increasing light absorptivity with higher thicknesses and a large thermal conductivity when thin films are being deployed. Lee et al. deposited a 100 nm thick reduced graphene oxide layer between a 500 μm thick Pyrex wafer and a 100 nm thick aluminum film, and a peak pressure of around 9 MPa was measured at 2.85 mm away from the coating using a 5 ns excitation laser with a laser fluence of 56 mJ/cm^2^, although the bandwidth was narrow [119]. More recently, Colchester et al. used an optical fiber coated with reduced graphene oxide and PDMS with a thickness under 50 μm to build an ultrasound transmitter with an outer diameter of 630 μm, which is illustrated in Figure 8 [118]. Besides the moderate biological safety, peak ultrasound pressures of 1.7 MPa were measured at 1.6 mm away from the transmitter, and a bandwidth of 24.3 MHz around the frequency of 14.7 MHz was perceived using an optical fluence of 15 mJ/cm^2^ and a pulse width of 2 ns.

The ability to optically generate ultrasound using graphite was demonstrated by Biagi et al. by covering the distal end of an optical fiber with a 20 μm thick mixture of graphite powder with epoxy resin [110]. When it was irradiated by a light source with a pulse duration of 6 ns, a pulse energy equal to 13 μJ and a repetition rate of 1.3 kHz, a peak-to-peak pressure of 20 kPa at a few centimeters from the probe, and a −3 dB bandwidth ranging frequencies from 10 to 40 MHz were recorded. The authors also emphasized the simple manufacture and miniaturization processes as well as the broad bandwidths achieved. However, the performance dependence on the concentration of graphite powder in the mixture and on the mixture layer thickness was acknowledged.

The integration of carbon nanoparticles is also easily reachable by using candle soot [120]. Additionally, candle soot nanoparticle–PDMS composite has shown great potential to photoacoustically generate high-intensity ultrasound waves in a wide frequency range mainly due to its large light absorption coefficient [120,121,122]. By irradiating a glass slide covered by a 5.99 μm thick candle soot nanoparticles-PDMS composite with a 3.57 mJ/cm^2^ laser energy density and a 6 ns pulse duration at a repetition rate of 10 Hz, a −6 dB frequency bandwidth of 21 MHz and a peak pressure of 4.8 MPa was found at 4.2 mm away from the transmitter [120,121]. Chang et al. measured a −6 dB frequency bandwidth of 22.8 MHz and a pressure of 3.78 MPa at a distance of 7.5 mm apart from a glass slide coated with a 2.15 μm thick candle soot-PDMS composite irradiated by a light source with the same pulse duration and repetition rate previously mentioned, but with a laser power of 1 mJ/cm^2^ [122].

Hsieh et al. deposited around 24.4 μm thick carbon nanofibers film between a glass substrate and a PDMS layer with a thickness of about 33.5 μm to optically generate ultrasound, as demonstrated in Figure 9 [123]. At 3.65 mm away from the transmitter, they measured a −6 dB bandwidth of 7.63 MHz and a mean peak pressure of 12.15 MPa when the absorption layer was irradiated by a laser source with a fluence of 3.71 mJ/cm^2^, a pulse duration of 4 ns, and a repetition frequency of 10 Hz.

Hou et al. covered a glass slide with an 11 μm thick carbon black-PDMS composite and irradiated it with an optical fluence of 0.03 J/cm^2^, a pulse duration of 5 ns, and a pulse repetition frequency of 5 kHz, obtaining a bandwidth above 40 MHz and a central frequency beyond 30 MHz [128]. Hsieh et al. also used a 30 μm thick layer of a composite of carbon black and PDMS to produce acoustic waves with a −6 dB bandwidth of 7.84 MHz and a peak pressure of 2.13 MPa at 3.65 mm away from the transducer by exciting the composite with a laser fluence of 3.71 mJ/cm^2^, a pulse duration of 4 ns, and a pulse repetition frequency of 10 Hz [123]. On the other hand, Buma et al. deposited a 25 μm thick film composed of a mixture of carbon black, PDMS, and toluene over a glass slide by spin coating, and when the compound was lightened by a laser with a 10 ns pulse length and a 30 nJ pulse energy, they reached −6 dB bandwidth of about 44 MHz around a central frequency of 30 MHz [82]. A carbon black spray paint was also used to cover the tip of the optical fiber [64]. A bandwidth over 20 MHz and a pressure around 70 kPa were measured at 2 mm away from the optical fiber tip when the optical source was a pulsed laser emitting pulses of 10 ns, at a rate of 1 kHz, and with a pulse energy of 8.6 μJ. Nevertheless, this carbon nanocomposite generally revealed narrower bandwidths and lower peak pressures than carbon nanofiber–PDMS, candle soot nanoparticle–PDMS, and chromium–PDMS composites [109,120].

CNTs are one of the most exploited carbonaceous materials, and some studies have already demonstrated their better performances than metal films [60,93,124] and gold nanocomposites [93]. This nanomaterial is an efficient photoacoustic generator since its heating process is almost immediate as a result of its great thermal conductivity and nanometric size [93,129,130]. Thus, the optoacoustic transient should be about the same as the laser pulse. Baac et al. confirmed the veracity of this hypothesis and demonstrated a better performance of CNT–PDMS composite when compared against a gold nanoparticles-PDMS composite and a chromium film, as can be seen in Figure 10. The similarity between the curve of CNT–PDMS and the laser one demonstrated the photoacoustic conversion independence of the frequency.

Ultrasound generation using CNTs was already explored by deposition onto concave lenses and glass surfaces. In [13], the authors used CNTs grown on a fused silica substrate and covered them with gold and a PDMS layer with a total thickness of 16 μm. They reported that when the 6 mm diameter lens was exposed to a laser fluence of 42.4 mJ/cm^2^ per pulse, a pulse duration of 6 ns, and a repetition rate of 20 Hz, frequencies over 15 MHz and peak pressures over 50 MPa could be obtained. Another study mentions using a lens with an aperture diameter of 15 mm covered with a CNTs-PDMS thin film and by applying an optical source with a fluence of 9.6 mJ/cm^2^ and a pulse duration of 6 ns, they obtained a −6 dB bandwidth of 25 MHz and a peak pressure of 70 MPa at a distance of 9.2 mm [61,124]. Lee et al. also deposited a CNTs-PDMS composite into the concave side of the fused silica lens with a 15 mm diameter [131]. Pressures over 30 MPa and a center frequency around 15 MHz were obtained by irradiating a pulsed laser with a pulse duration of 6 ns and pulse energies of 14, 16, 17.5, and 18.5 mJ. Another study reported that when a 49 μm thick layer of multi-walled carbon nanotubes (MWCNTs) and PDMS was deposited onto a glass slide with a polyimide film coverage and illuminated by a pulsed laser with a pulse width under 5 ns, pulse energy of 76 μJ, and a repetition rate of 2 kHz, a bandwidth of around 27.1 MHz and an acoustic pressure of 0.977 MPa were recorded 2.7 mm away from the transmitter [65].

Most recently, several studies analyzed the ability of CNTs to optoacoustically generate ultrasound when deposited onto optical fiber tips. Colchester et al. covered an optical fiber tip with a 10 μm thick CNTs-PDMS layer [83]. By using two different optical fibers with a core diameter of 105 μm under laser fluence of 41.6 mJ/cm^2^ and 200 μm with an optical fluence of 36.3 mJ/cm^2^, both with 2 ns pulses at a repetition rate of 1 kHz, the −6 dB bandwidths were 12 MHz, and 15 MHz and the peak pressures at the end face of the fiber were 3.6 and 4.5 MPa, respectively. In another study, to make an ultrasound probe with a diameter under 0.84 mm, an optical fiber was covered with a mixture of functionalized CNTs, xylene, and PDMS under pulsed laser excitation with pulse width of 2 ns, a repetition rate of 100 Hz and an optical fluence of 96.1 mJ/cm^2^, and the peak pressure and bandwidth measured at end face of the optical fiber were 4 MPa and around 20 MHz, respectively [60]. Poduval et al. coated the distal end of an optical fiber with a 13.7 μm thick layer of multi-walled carbon nanotubes (MWCNTs) by electrospinning and a PDMS layer by dip-coating [126]. They mentioned that when a laser with 2 ns long pulses at a repetition rate of 100 Hz and a laser fluence of 35 mJ/cm^2^ were applied, a peak-to-peak pressure of 1.59 MPa at 1.5 mm away from the probe and a −6 dB bandwidth of 29 MHz around the frequency of 31 MHz were achieved. Later, Colchester et al. also covered a fiber tip with a composite of MWCNTs and PDMS to construct an ultrasound probe with an outer diameter of 1.25 mm [1]. Through the excitation with 2 ns wide pulses at a repetition rate of 8 kHz and a pulse energy of 40 μJ, they perceived a peak-to-peak pressure of 1.87 MPa at 1.5 mm from the end of the fiber and a −6 dB bandwidth of 31.3 MHz. Additionally, Finlay et al. reported a peak-to-peak pressure of 8.8 MPa at 1.5 mm away from the probe and a −6 dB bandwidth of 26.5 MHz by exploiting a similar optical fiber coating and irradiating with a laser with pulse energy of 20 μJ, a pulse duration of 2 ns, and a repetition rate of 50 Hz [59]. Noimark et al. also assessed the performance of ultrasound probes comprising optical fibers coated with MWCNT–PDMS composites but constructed through different fabrication processes [63]. The application of an optical excitation with a pulse width of 2 ns and a fluence of 33.1 mJ/cm^2^ resulted in the performance summarized in Figure 11.

Although randomly oriented CNTs have been used in most studies, which makes the material possess isotropic properties on average that degrade its performance, the possibility of producing ultrasound by photoacoustic effect, using arrays of vertically aligned CNTs, commonly designated CNTs forests, has been examined. As reviewed in [129], such an arrangement of CNTs can lead to an enhancement in both performance and reproducibility. This is mainly due to the higher thermal conductivity coefficient along the length direction when compared to the diameter direction [127]. Tahmid et al. also demonstrated that the light absorbance is maximized when the direction of light incidence is equal to the length direction of the CNT, as shown in Figure 12 [132].

#### 3.2.5. Comparison between Optically Absorbing Materials

Depending on the laser wavelength available to develop a US probe and the acoustic frequency range that is intended, it may be more convenient to use a particular substance as the photoacoustic target rather than another. Figure 13 depicts the normalized power spectra of the abovementioned materials employed for photoacoustic-based ultrasound probes, and Figure 14 displays the wavelength at which some of these materials have the maximum optical absorption. It is relevant to emphasize that the latter graph does not allow a comparison between the different materials since the curves were originally in different units, but it provides valuable information about the optimal spectral regions of operation of these materials. These graphs show not only that considerable acoustic power is generated over a wide frequency range for most materials but also that the optimum laser wavelength varies greatly depending on the material in use. Furthermore, most materials show preferential absorption in the ultraviolet region, with the exception of gold nanoparticles and crystal violet, which absorb mostly in the visible region of the electromagnetic spectrum, and single-walled carbon nanotubes, which absorb preferentially in the near-infrared range. Based on this information, one can then choose the material that will render the best performance under the setup conditions, also taking into consideration whether the required technology to manufacture it is owned and mastered.

As evidenced before, a wide variety of materials have already been used to optoacoustically generate ultrasound. However, to further performance enhancement, new materials are still under study. Photostable dye-PDMS composites with a thickness under 20 μm were also deposited onto optical fiber tips to generate ultrasound through a photoacoustic effect [133]. Peak-to-peak pressures over 2 MPa and bandwidths near 30 MHz were measured when the laser had a pulse width of 2 ns, a repetition rate of 100 Hz, and a pulse energy of 20.1 μJ. In addition, mixtures of quantum dots with PDMS [61,140] have also been researched due to their good photostability and adjustable optical absorption profiles leading to high absorbance at the intended wavelengths, as well as plasmonic structures, namely Tamm plasmon structures [141] that reach total absorption of laser pulses irrespective of wavelength.

### 3.3. Acoustic Energy Release Materials for Ultrasound Probes

The use of elastomeric materials is an effective solution for poor thermal expansion coefficients [60,61,62,63,75], increasing the conversion efficiency by over 20 dB when compared with the performance of thin metallic films [82]. Some studies confirmed the enhanced performance using epoxy [62,110], Parylene [61] and PDMS [61,63,93,116,118]. For example, Lee and Guo sandwiched thin chromium and titanium films between elastomers with a top layer of aluminum, as displayed in Figure 15, and verified the augmented performance, achieving transmitted ultrasound pressures of up to 1.82 MPa when the chromium structure was irradiated by a laser fluence of 2.35 mJ/cm^2^ with 6 ns wide pulses [142].

These elastomeric materials are typically deposited onto the optical absorber material. The PDMS layer is usually built up through dip-coating [126] or spin-coating [120], but it can also be mixed with other substances [116]. Parylene coatings were applied using physical vapor deposition [61], while the epoxy was mostly mixed with the optical absorbers [62].

PDMS has already demonstrated greater photoacoustic efficiency than both epoxy in [62] and Parylene in [61], and it has a thermal expansion coefficient over an order of magnitude greater than other metals, resulting in higher pressures and conversion efficiencies [61,82,93]. Furthermore, this elastomer seems suitable for biomedical applications since it is biologically safe and its acoustic impedance is comparable to biological tissue, which prompts an efficient coupling between the probe and the tissue [61,82]. Another great feature of PDMS is the possibility of its manipulation at a micrometer scale, enabling it to make devices with a high degree of miniaturization [61].

The undeniable virtues of PDMS make it very promising for integration with optical absorbers [75], such as organic pigments (e.g., crystal violet) [61,75], metallic nanoparticles (e.g., gold nanoparticles) [61,75,109,115], and carbonaceous materials [61,63,118,123]. Its advantageous properties explain why PDMS is more often employed than other elastomers, as could be seen throughout the previous subsection.

## 4. Modeling and Design

One key step to obtaining an ultrasound probe with the required features is the probe design, where material properties and dimensions are selected. It can be performed experimentally or based on equations and computational tools. Experimental parametric analysis and measurements can be highly time-consuming and costly [143]. Therefore, a key step when developing a system based on the photoacoustic effect is to implement a multiphysics model for this phenomenon in order to enable the analysis of several phenomena, geometries, and materials and understand how each of the variables and characteristics will contribute to the overall performance of the system.

This can be accomplished by mathematically modeling this effect through extensive and detailed analysis of a set of equations that describe the manifold phenomena concerned. This approach has already been used successfully in [30] and in [144]. However, this process can be quite cumbersome, more susceptible to small errors, and can be quite challenging, even without closed-form solutions, when complex, realistic probe geometries are involved.

As an alternative to mathematical modeling, based on closed-form equation solving, the use of the k-wave toolbox, which is an open-source acoustics toolbox for MATLAB, is quite recurrent for photoacoustic effect modeling [145]. Both Mastanduno et al. [146] and Agrawal et al. [147] used this toolbox together with the NIRFast package, which allows the modeling of near-infrared light transport in tissue. Conversely, other authors have used this toolbox along with Monte Carlo modeling and achieved great accuracy [148,149,150], although this approach is commonly described as computationally demanding.

In addition, computational models can be developed in multiphysics simulation tools. This simple yet suitable method provides pretty accurate predictions of the real-world setups’ performance and processes efficiently and cheaply through a virtual environment [83,148]. These simulation models require validation against experimental results notwithstanding, they provide a prediction of system performance and allow several output variables to be easily analyzed, even when more intricate geometries are involved. Similarly to any information that is digitally processed, data must also be discretized in computational models. The finite difference method, the finite element method and the finite volume method are the three main discretization methods, but only the latter two can be applied to any geometries [151].

The use of a commercial multiphysics simulation tool is another alternative to model and assess the performance of a system that relies on the photoacoustic effect [92,152,153,154,155]. Figure 16 illustrates the simulation results of the photoacoustic effect using one of the commercial tools, compared to the analytical results, revealing consistency between these two analysis approaches [154].

Such software tool has a wide diversity of modules available that allow solving problems involving different physics and that can be described by partial differential equations, already embedded in the tools or added by the designer in case de phenomena is not modeled by the tool. This allows us to simulate this effect very accurately on a wide range of materials and geometries.

### Main Steps for Modeling Using a Commercial Tool

When modeling a particular phenomenon using these tools, one should try to replicate the real conditions as closely as possible yet counterbalance this so that the model does not become too complex and computationally burdensome. To this end, it is often desirable to introduce some simplifications. The first step when modeling any physics phenomenon in commercial multiphysics simulation software consists of building or importing the geometry to be modeled, as exemplified in Figure 17. In this figure, the two gray blocks together are the photoacoustic target, and the blocks in shades of red are the media where the generated acoustic waves will propagate.

Each domain of the designed geometry must also be assigned the corresponding material. Lots of different materials are available in the simulation tool’s libraries, but a new material can also be manually configured. Subsequently, it should be assigned the physics and multiphysics modules according to the phenomena concerned. To avoid having to use the electromagnetic module, for simplicity, the following equation describing the heating of the photoacoustic target following laser incidence can be included:q = (1 − R) I_0_ α,(1)
where I_0_ corresponds to the laser intensity, R to the reflectivity, and α to the optical absorption coefficient of the photoacoustic target material [139]. By using this expression, the use of additional physics is spared, which makes the problem less complex and quicker to solve. As a result, it is only required to add physics and multiphysics capable of simulating the thermal expansion arising as a consequence of the laser incidence and the subsequent generation of acoustic waves, as well as their propagation throughout a given medium. It is then important to specify the boundary conditions appropriate to the problem at hand and depending on the factors that are to be disregarded or not. These conditions have to be carefully defined so that a solution to the problem can be found since troubles experienced during simulation modeling are often related to missing or incorrectly set boundary conditions.

Afterward, an appropriate mesh must be created. The mesh defines how many nodes the simulation model will have to predict the output properties, and each element of the mesh has to be smaller than the smallest dimension of the selected domain. The more refined the mesh, the more accurate the results, although it takes longer to find the solution. Consequently, the mesh should only be refined in areas where there are more details or smaller regions. Figure 18 depicts a mesh suitable for the previous geometry, where a more refined mesh is present only in those regions where structures with reduced dimensions can be found.

Lastly, the studies and analyses intended to be performed must be specified according to the parameters sought to be analyzed and with the appropriate temporal and/or spectral resolution. Following the aforementioned steps, a model of the photoacoustic effect can be successfully built, and a wealth of results can be gathered and scrutinized, as displayed in Figure 19. Information regarding, for instance, temperature and displacement magnitude can also be provided.

Noticeably, these tools usually allow results to be obtained and analyzed both over time and frequency. In this way, pretty relevant information for assessing such a system can be gleaned, and also allow conclusions to be drawn concerning the devices’ requirements that should be used when experimentally implementing a system relying on the photoacoustic effect.

## 5. Characterization

After having the probe available, the last step is to characterize its performance. That requires a setup based on optical sources and acoustic detectors (optical or not). This section delves into a review of several experimental setups employed for ultrasound generation through the photoacoustic effect, identifying different optical excitation sources, methods for ultrasound beam steering, and ultrasound detectors already explored.

The advantages of using optical fibers as optimal light delivers are broadly recognized [156,157]. In addition, besides the high degree of miniaturization, optical fibers also offer great flexibility that could be convenient in the medical field, especially in minimally invasive procedures [63,133]. As such, they are exploited commonly in photoacoustic-based systems [74]. Figure 20 depicts an example of an all-optical experimental setup for a photoacoustic-based system, which therefore comprises optical fibers both for ultrasound emission and detection. Nevertheless, there are also experimental setups where optical fibers are employed only as a part of the ultrasound transmitter and the receptor is a commercial hydrophone [76].

### 5.1. Photoacoustic Emission

The properties of the optical excitation source severely influence the properties of the generated acoustic wave. Firstly, the laser wavelength should be selected according to the maximum absorption wavelength so that the photoacoustic material reaches high photoacoustic efficiency [62,92]. In addition, the light pulse duration affects the range of ultrasound frequencies [1], and by increasing the laser intensity and decreasing the beam diameter, higher laser fluences are obtained and, consequently, greater acoustic power [61]. Furthermore, as previously mentioned, controlling the photoacoustic target materials’ thickness is also crucial for achieving broad bandwidths and large acoustic pressures [63,83].

Table 1 displays a summary of different optical sources already employed in photoacoustic systems, the material used as a photoacoustic target, and the resulting performance. As can be seen, a pulsed laser in the visible or infrared spectral range is usually the optical source in photoacoustic systems [6,8,9,49]. It is also important to highlight that pulsed lasers are always preferable to continuous waves because they provide an enhanced signal-to-noise ratio, improving image quality [54].

Although good performance is achieved with pulsed lasers, these optical sources are quite expensive and bulky [21]. Consequently, other optical sources are being explored held with the aim of constructing handheld probes. The feasibility of using laser diodes and light emitting diodes (LEDs) for clinical imaging was already demonstrated [21,71], although poor imaging speed and limited output power, even when combined in arrays, were commonly reported [21,49,57]. Another alternative is the use of xenon flash lamps that are cheap and safe, regardless of their low operating frequencies [159].

Once the optical source has been chosen, the light needs to be guided toward the target material. Light is typically delivered through optical fibers, and the coupling with the light source could be achieved using a coupler [62,116,124,160] (e.g., F810SMA-543 and DC1300LEFA from Thorlabs), a collimator [7,30,60,74,161] or both a collimator and convex lens [128].

### 5.2. Ultrasound Beam Steering

The scan of the transmitted ultrasound beam is essential to later reconstruct a 2D or 3D image. For intravascular monitoring, it may be necessary to perform a 360° scan to observe the entire vessel wall. Rotary systems that include stepper motors are the typical scanning method in these cases [1]. However, for most applications, a flat scan is desired to obtain a frontal view. Two-axis or three-axis translation stages are usually engaged to perform linear scans [60,62,63,64,83,142]. On the other hand, in [160] a raster scanning method was implemented through a voice coil stage and a linear motor. Additionally, Lee et al. constructed a scanner based on the Scotch yoke mechanism, which converts the linear motion of a slider into rotational motion, or the other way around [162]. In spite of the reduced dimensions of most of these motorized stages (on the order of tens/hundreds of millimeters), they are still too large to be incorporated into highly miniaturized probes, and their scanning speed is reduced. These limitations can be overcome through the use of microelectromechanical systems (MEMS) moving platforms, such as those pictured in Figure 21, that have their dimensions extremely reduced, which is favorable for integration in miniature ultrasound probes [163,164,165,166,167,168].

Nevertheless, MEMS platforms generally exhibit reduced scanning speed and stability [37]. Galvanometer-based scanning methods can be used instead. Alternatively, polygon-mirror scanners and microlens arrays can also be employed since they can both perform high-speed scanning and the latter with the additional benefit of not needing to perform mechanical scanning [37].

### 5.3. Ultrasound Detectors

The photoacoustic imaging system performance is heavily dependent on the proper choice of the US detector, which should have a center frequency matching the center frequency of the photoacoustic signal, and the broadest bandwidth possible to provide sufficient spatial resolution [7,57]. Piezoelectric detectors, such as lead zirconate titanate (PZT), lithium niobate, and mainly polyvinylidene fluoride (PVDF), are frequently included in photoacoustic systems as ultrasound detectors due to their large bandwidth [57,82,110,142], even though typical low sensitivities, especially for highly miniaturized devices [57,75]. Table 2 shows some piezoelectric-based hydrophones already employed in photoacoustic systems.

CMUTs are another possible ultrasound detector that could be included as they are highly miniaturized [57]. However, as mentioned before, their reduced performance and high cost are their main drawback [60,88,89,90].

Conversely, optical ultrasound detection holds great potential for building miniaturized photoacoustic systems, as they are associated with broad bandwidths that enhance spatial resolution, and they are able to preserve their sensitivity even when miniaturized to a micrometer scale [75]. Several optical ultrasound detectors, such as those illustrated in Figure 22, were already incorporated into photoacoustic systems or simply for ultrasound detection.

For instance, the Mach-Zehnder interferometer had the drawback of only being able to image objects smaller than the detector [175,176,177,178], fiber Bragg grating’s sensitivity does not depend on transducer size, and it is cheap [173,179,180,181,182], micro-ring resonator presented large detection bandwidth [174,183,184,185,186], and Fabry-Pérot sensor is the most optical ultrasound detector exploited due to its suitable performance [1,63,128,160,187,188,189,190]. Table 3 describes the performance of photoacoustic-based ultrasound probes that included Fabry-Pérot as ultrasound transducers. However, the main weaknesses of optical ultrasound detectors lie in the increased complexity and delicateness regarding instrumentation [120].

After the ultrasound detection, the signal may need to be amplified and then displayed on an oscilloscope. Postprocessing algorithms may also be required to eliminate unwanted effects or artifacts to further 2D or 3D image reconstruction as described in [1,59,60,63].

## 6. Outlook and Challenges

The broad use and acceptance of devices relying on the photoacoustic effect, as well as the growth of this imaging modality in a research setting, are clear. Despite the growing interest, after going through all the relevant aspects that require consideration when evaluating the use of photoacoustics as a method for non-destructive characterization, it is found that there are still some remaining challenges that are regularly addressed and in need of careful understanding prior to its selection.

The first challenge when deciding to use the photoacoustic effect to obtain an ultrasound probe is the material selection. Such materials must be selected to achieve high photoacoustic effect efficiency but should be evaluated tradeoffs with other aspects. Tradeoffs with the laser wavelength required for excitation, which may have an impact on laser source requirements and cost, must be considered. In addition, tradeoffs with the fabrication methodology must be counterbalanced as some materials may undergo a very complex fabrication and/or deposition process. Finally, tradeoffs with the required miniaturization must also be taken into account, as when a high degree of miniaturization is a requirement, there may also be some hurdles in controlling the dimensions and homogeneity achieved.

After finding the convenient material(s), modeling the full system can be very complex if a detailed and precise model is required to deliver accurate simulated results. Since this is a phenomenon that entails several domains of physics, it becomes a laborious problem to solve, particularly when intricate geometries are at issue. In an attempt to overcome this pitfall, computational tools are usually employed. Even though the use of simulation software is a great support, sometimes the problem may get computationally heavy, which can hinder solving the problem in a timely manner. On top of complex geometries and multiple domains, such simulations also involve a broad range of timescales. Even considering that the light wave is assumed as a pulse in the nanoseconds range, the generated ultrasound wave will be in the mili- or microsecond range. Additionally, since the light-absorbing particles are in the nanoscale (to interact with light), and the full device is in the mili- or microscale, the model will comprise one or two order(s) of magnitude. This leads to a mesh challenge, as well as large computational times.

The final challenge will be the performance assessment, which entails the accurate measurement of the pressure and frequency generated by the probe in the function of a precise light beam, carefully switched. Even when using advanced equipment that allows the control of each parameter at the system output and the detector input, it may be hard to accurately estimate the exact value of some variables. For example, the power, or the intensity, of the beam that reaches the photoacoustic target may be difficult to assess due to attenuations that occur in the path as a consequence of light interaction with air particles or containers where the experiments will be carried out. Furthermore, even the frequency may be difficult to assess due to multiple reflections of ultrasound waves along its propagation path. In addition, acoustic receivers should be broadband and equally sensitive over a wide-angle span, for more accurate detection of the acoustic wave that was generated, due to sometimes generating an ultrasound wave that may be possible to determine only in some range, and the acoustic properties are not fully known beforehand with the required details for the models used in the simulation.

Even though, as previously discussed, there are still several open issues requiring attention when considering the use of photoacoustic non-destructive analysis, such modality shows to be quite a viable and promising solution that holds great potential, predominantly for biomedical applications.

## Figures and Tables

**Figure 1 sensors-22-09541-f001:**
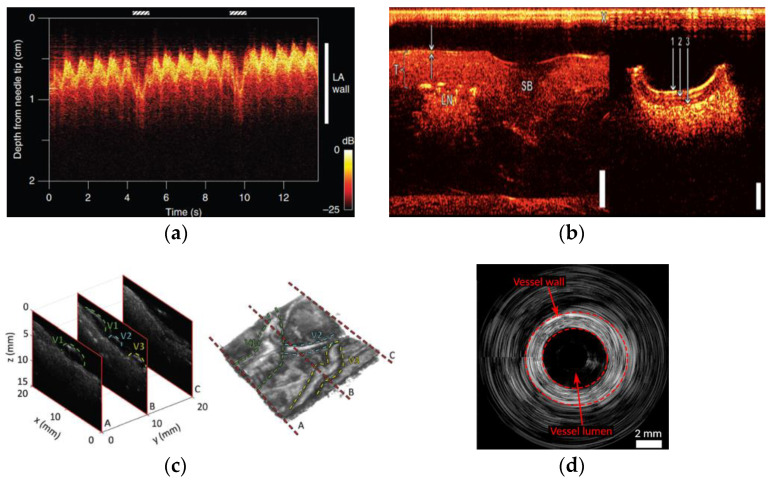
Images generated with photoacoustic-based ultrasound probes. (**a**) Left atrium (LA) wall M-mode image of a swine’s heart; (**b**) 2D images of aorta (left) and carotid artery (right) of swine samples (scale bar: 2 mm); (**c**) 2D images of an ex vivo piece of normal term human placenta (left) and 3D rendering of the reconstructed image (right); (**d**) B-mode intraluminal imaging of a swine carotid artery. Reproduced with permission from [1,59,60,61].

**Figure 2 sensors-22-09541-f002:**
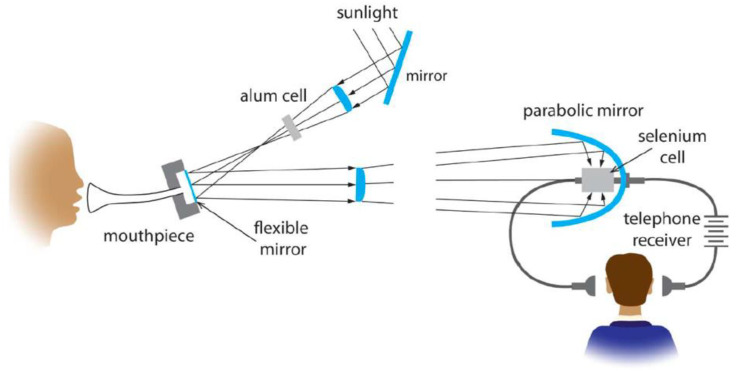
Scheme of the photophone setup, the first optical communication device created by Alexander Graham Bell and his assistant. Reproduced with permission from [68].

**Figure 3 sensors-22-09541-f003:**
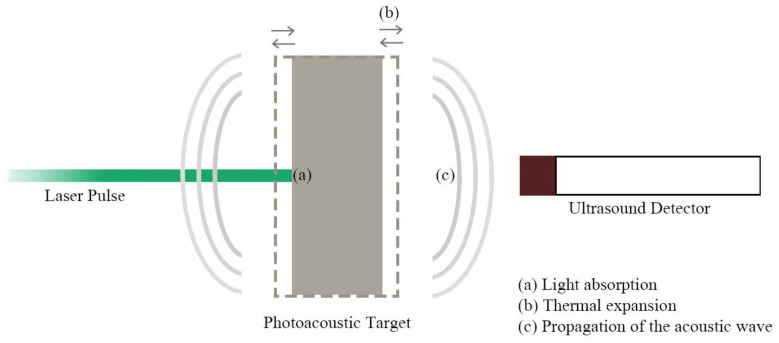
Schematic representation of the photoacoustics effect.

**Figure 4 sensors-22-09541-f004:**
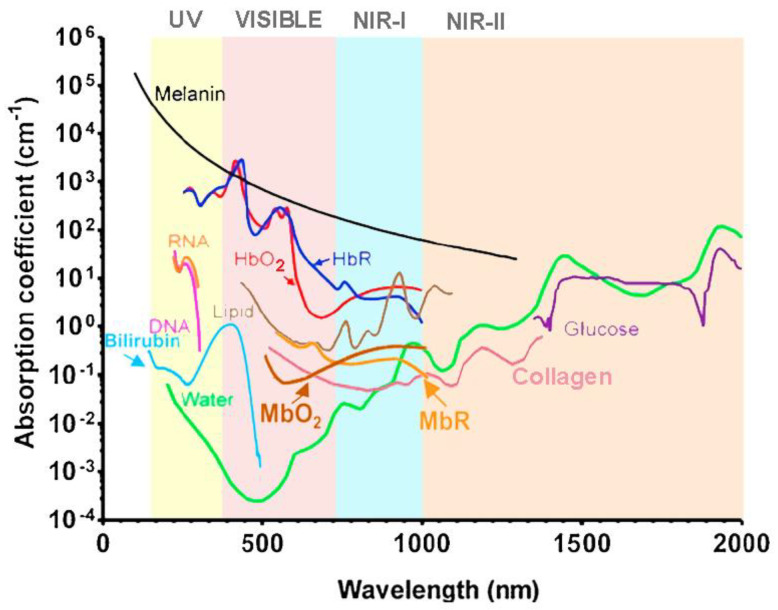
Absorption coefficient versus wavelength for different endogenous contrast agents. Adapted from [12,23,30,44].

**Figure 5 sensors-22-09541-f005:**
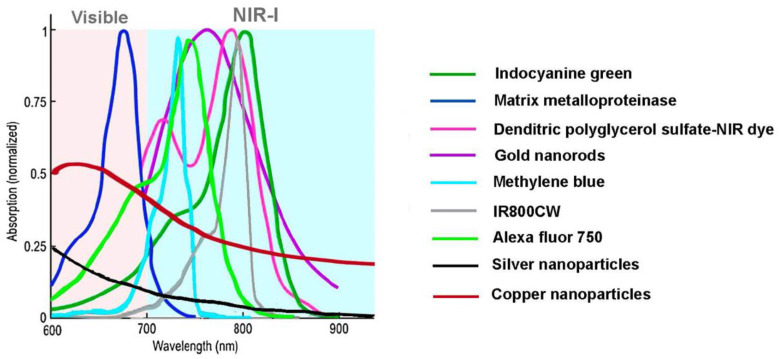
Normalized absorption coefficient versus wavelength for some exogenous contrast agents. Adapted from [23,78,107,108].

**Figure 6 sensors-22-09541-f006:**
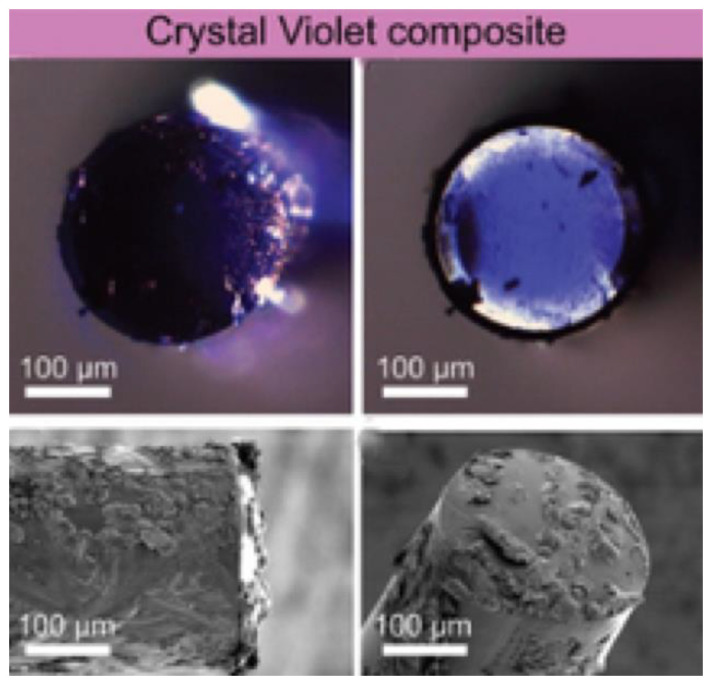
Images of the optical fiber’s distal end covered by crystal violet-PDMS composite. Reproduced with permission from [61].

**Figure 7 sensors-22-09541-f007:**
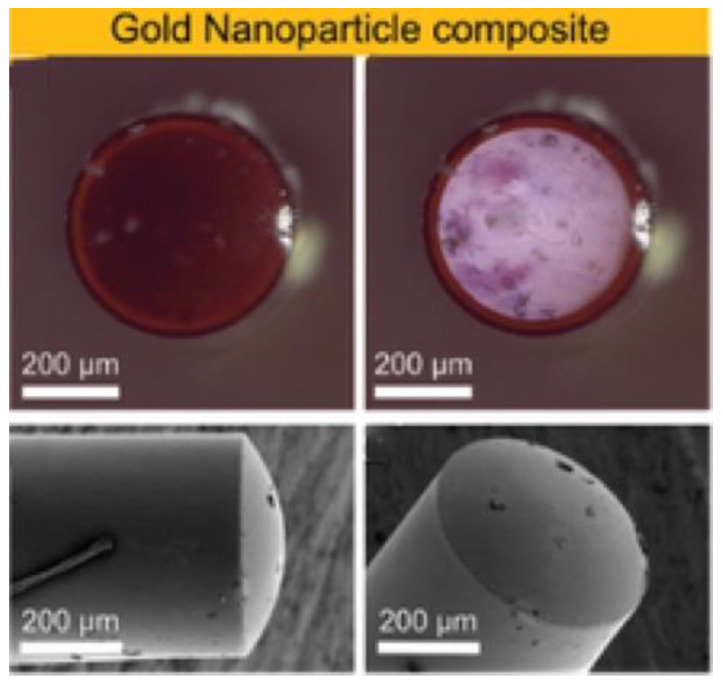
Images of the optical fiber’s distal end covered by gold nanoparticles-PDMS composite. Reproduced with permission from [61].

**Figure 8 sensors-22-09541-f008:**
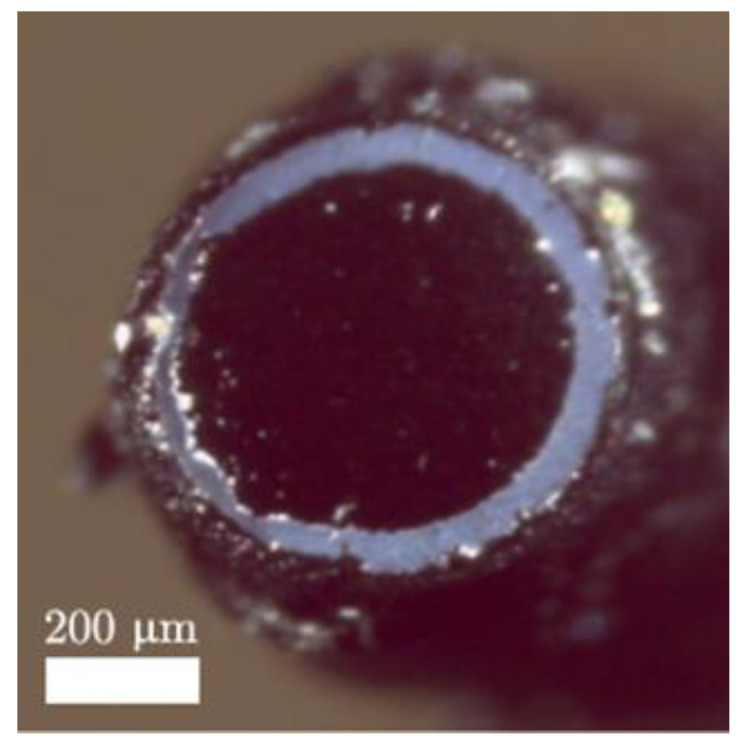
Image of the optical fiber’s distal end covered by reduced graphene oxide combined with PDMS. Reproduced with permission from [118].

**Figure 9 sensors-22-09541-f009:**
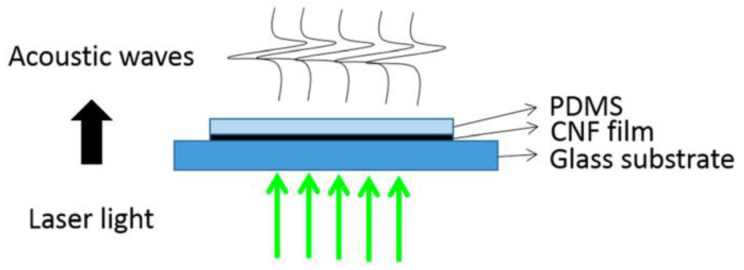
Organization of the carbon nanofibers and PDMS layers. Reproduced with permission from [123].

**Figure 10 sensors-22-09541-f010:**
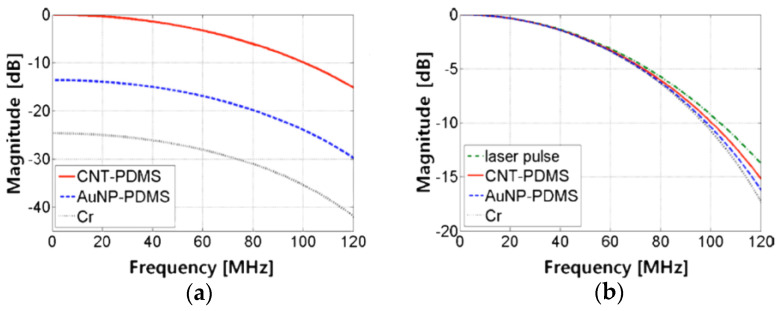
Comparison between the performance of CNT–PDMS composite, gold nanoparticles-PDMS composite, and chromium film. (**a**) Frequency spectra normalized to the DC value of the CNT–PDMS composite. (**b**) Frequency spectra normalized to each DC value compared to that of the laser that was employed. Reproduced with permission from [93].

**Figure 11 sensors-22-09541-f011:**
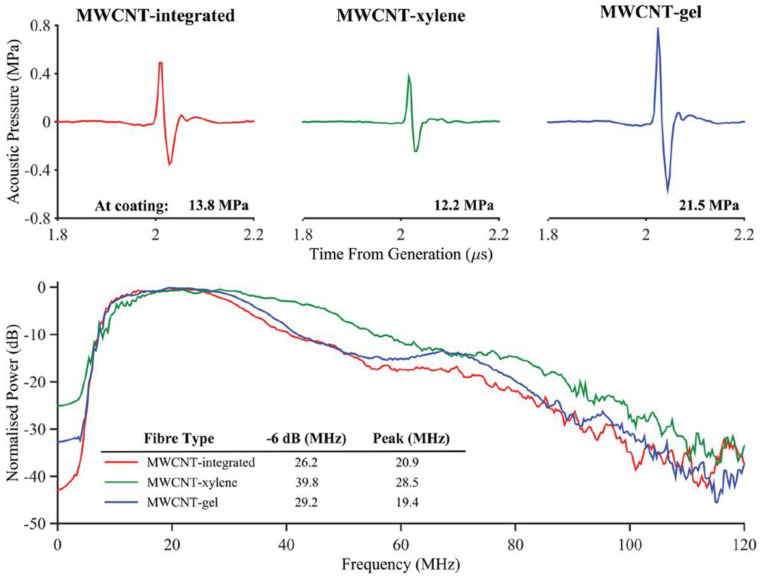
Comparison of acoustic pressure at 3 mm away from the coatings and normalized power spectra generated for the MWCNT–PDMS integrated coating (red line), MWCNT–xylene/PDMS coating (green line), and MWCNT–gel/PDMS coating (blue line). Reproduced with permission from [63].

**Figure 12 sensors-22-09541-f012:**
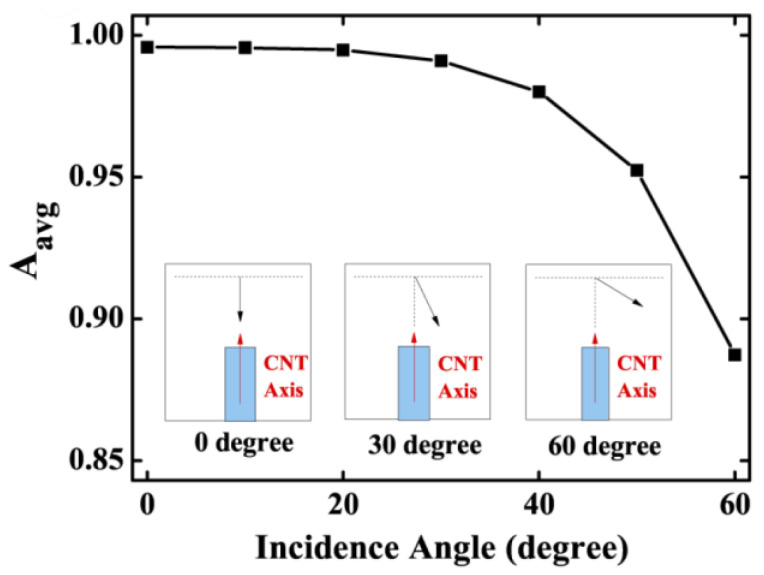
CNTs’ average light absorbance (A_avg_) as a function of the light incidence angle. Reproduced with permission from [132].

**Figure 13 sensors-22-09541-f013:**
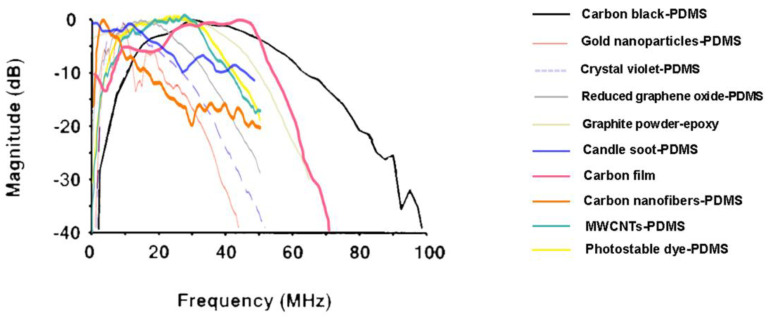
Normalized ultrasound power spectra of several materials used for photoacoustic-based ultrasound transmitters. Adapted from [61,82,110,118,120,123,125,126,133].

**Figure 14 sensors-22-09541-f014:**
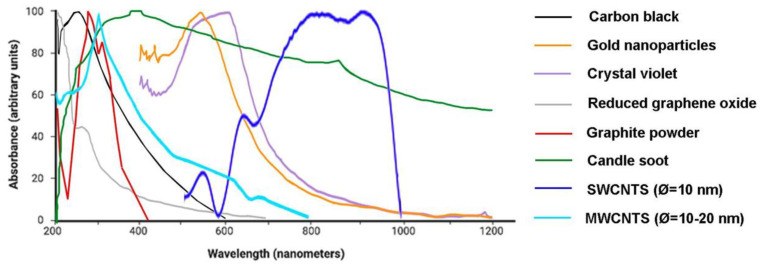
Curves showing the wavelength at which maximum optical absorption occurs for each material. Adapted from [61,134,135,136,137,138,139].

**Figure 15 sensors-22-09541-f015:**
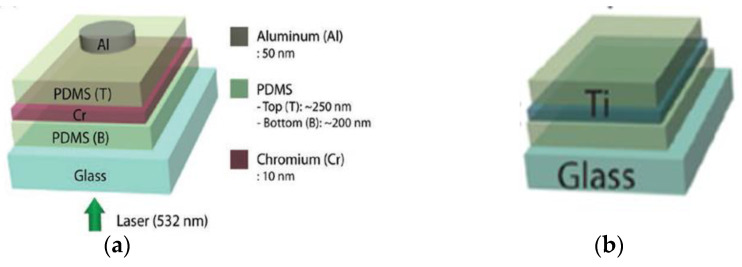
Combination of PDMS layers with thin metallic layers. (**a**) Chromium layer between two PDMS layers. (**b**) Titanium layer sandwiched between PDMS layers. Reproduced with permission from [142].

**Figure 16 sensors-22-09541-f016:**
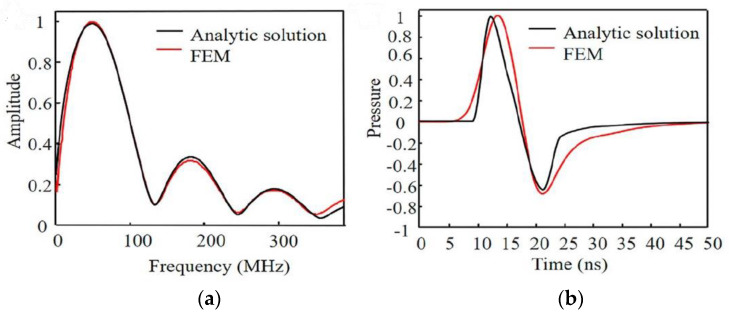
Ultrasound wave generated through the photoacoustic effect in a model developed in COMSOL Multiphysics compared against an analytical solution: (**a**) over time; (**b**) over frequency. Reproduced with permission from [154].

**Figure 17 sensors-22-09541-f017:**
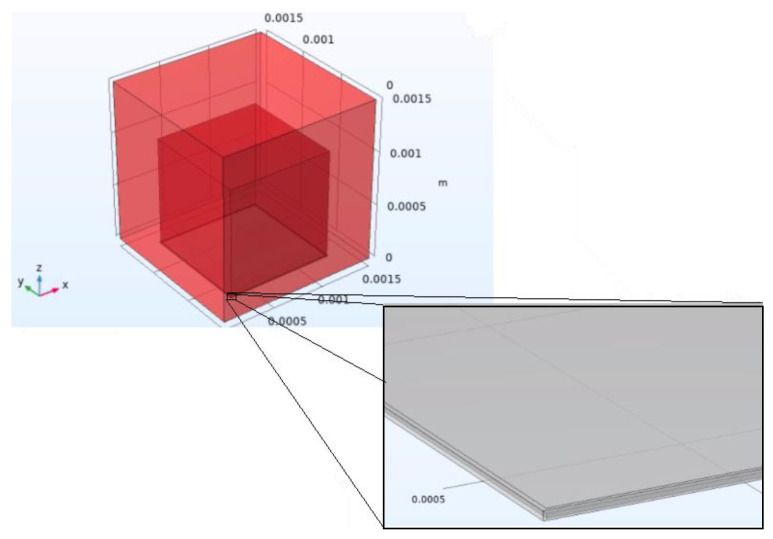
Example of geometry designed in a simulation tool.

**Figure 18 sensors-22-09541-f018:**
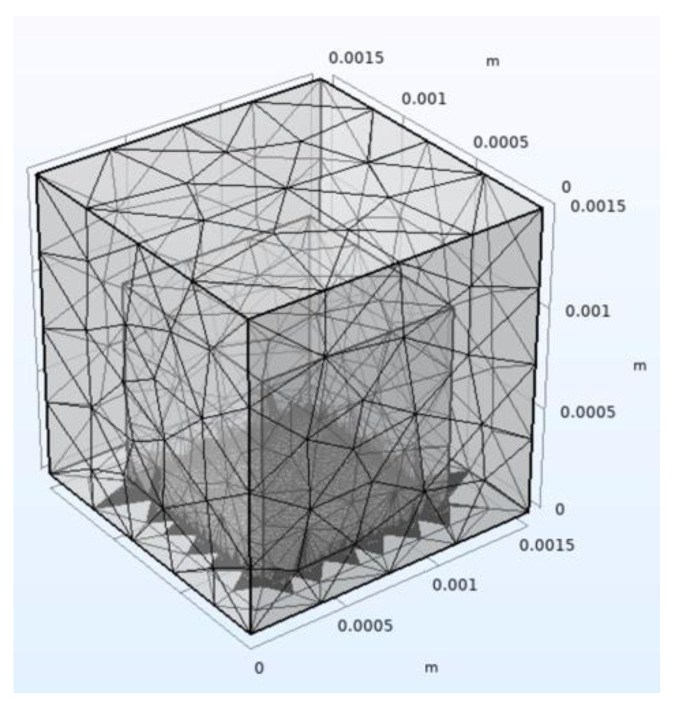
Mesh built in a simulation tool.

**Figure 19 sensors-22-09541-f019:**
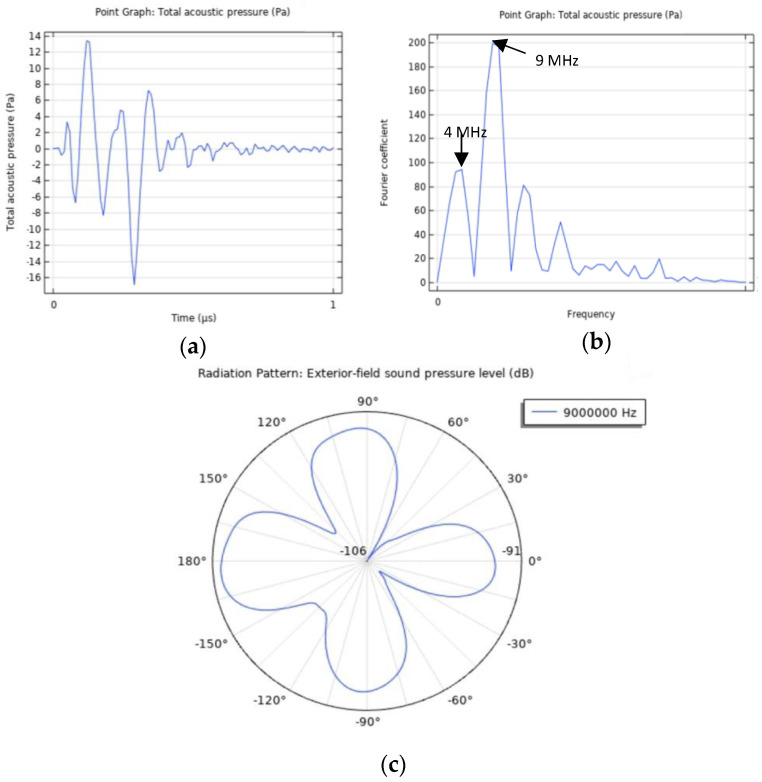
Photoacoustic effect simulation results. (**a**) Acoustic pressure over time. (**b**) Acoustic pressure spectrum. (**c**) Radiation pattern for the predominant emission frequency.

**Figure 20 sensors-22-09541-f020:**
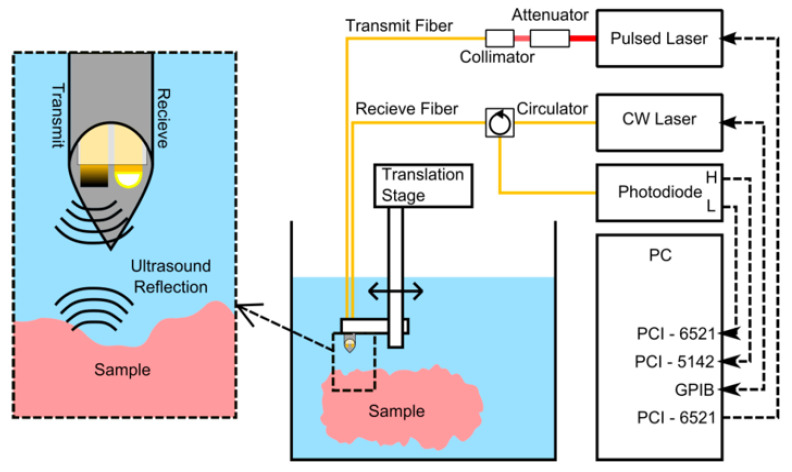
Schematic of the experimental setup for an all-optical ultrasound probe. Reproduced with permission from [60].

**Figure 21 sensors-22-09541-f021:**
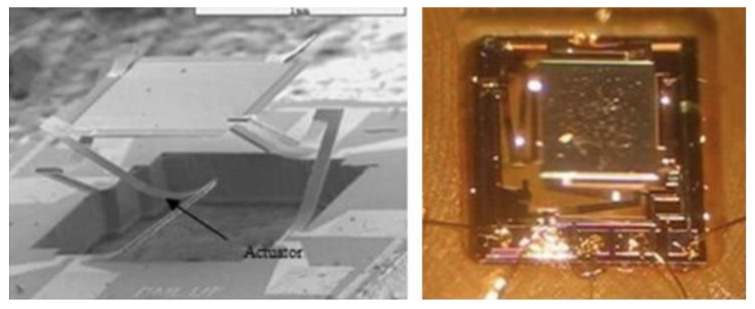
Pictures of MEMS moving mirrors. Reproduced with permission from [167].

**Figure 22 sensors-22-09541-f022:**
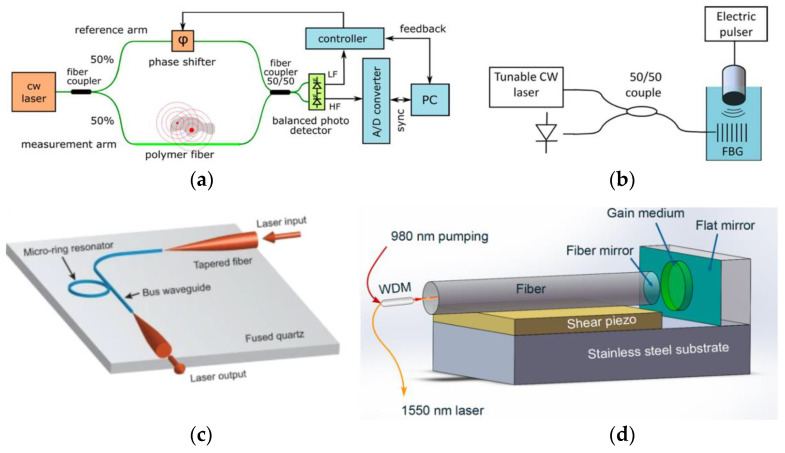
Schematic of optical ultrasound detectors. (**a**) Mach-Zehnder Interferometer. (**b**) Fiber Bragg grating. (**c**) Micro-ring Resonator. (**d**) Fabry-Pérot. Reproduced with permission from [91,173,174,175].

**Table 1 sensors-22-09541-t001:** Summary of the laser properties employed in photoacoustic systems, including different materials and the consequent US features.

Refs	Material (Thickness)	Laser model and Manufacturer	Wavelength [nm]	Repetition Rate [Hz]	Pulse Width [ns]	Laser Fluence/ Energy	Bandwidth [MHz]	Distance [mm]	Acoustic Pressure [MPa]
[142]	PDMS (200 nm)–Cr (10 nm)–PDMS (200 nm)–Al (50 nm)	Surelite I-20, Continuum	532	-	6	2.35 mJ/m^2^	-	-	1.82
[61]	Crystal violet–PDMS (20 μm)	-	532	100	-	86.3 mJ/cm^2^	15.1 (−6 dB)	1.5	0.90 (peak-to-peak)
[109]	Gold nanoparticles–PDMS (~4.5 μm)	Surelite, with OPO Plus, Continuum,	700	-	5	100 mJ	65	-	1.5
[62]	Gold nanoparticles–PDMS (105 μm at maximum)	Surelite-I-10, Continuum	532	10	5	8.75 mJ/cm^2^	>20	1	0.64 (peak-to-peak)
[116]	Gold nanoparticles–PDMS (450 μm)	Falcon 527-30-M, Quantronix	527	1000	150	13 mJ/cm^2^	3.1	1.8	0.18949
[61]	Gold nanoparticles–PDMS (200 μm)	-	532	-	-	55.3 mJ/cm^2^	4.5 (−6 dB)	1.5	0.41 (peak-to-peak)
[118]	Reduced graphene oxide-PDMS (~50 μm)	SPOT-10–500-1064, Elforlight	1064	100	2	15 mJ/cm^2^	24.3	1.6	1.7 (peak)
[119]	Pyrex (500 μm)–reduced graphene oxide (100 nm)–aluminum (100 nm)	Surelite, Continuum	532	-	5	56 mJ/cm^2^	(narrow)	2.85	~9 (peak)
[110]	Graphite powder and epoxy resin mixture (20 μm)	LCS-DTL-122QT, Lasertech	1064	1300	6	13 μJ	30 (−3 dB)	few cm	0.020 (peak-to-peak)
[120]	Candle soot nanoparticles–PDMS (5.99 μm)	SL-III-10, Continuum	532	10	6	3.57 mJ/cm^2^	21 (−6 dB)	4.2	4.8 (peak)
[122]	Candle soot–PDMS (2.15 μm)	SL-III-10, Continuum	532	10	6	1 mJ/cm^2^	22.8 (−6 dB)	7.5	3.78
[123]	Carbon nanofibers (24.4 μm)– PDMS (33.5 μm)	Minilite I, Continuum	532	10	4	3.71 mJ/cm^2^	7.63 (−6 dB)	3.65	12.15 (peak)
[64]	Carbon black spray paint	FQ-200-20-V-532, Elforlight	532	1000	10	8.6 μJ	20	2	0.070
[128]	Carbon black–PDMS (11 μm)	MIRVISION, Keopsys	1064	5000	5	0.03 J/cm^2^	41 (−6 dB)	-	-
[82]	Carbon black–PDMS (25 μm)	-	-	-	10	30 nJ	44 (−6 dB)	-	-
[123]	Carbon black (30 μm)–PDMS	Minilite I, Continuum	532	10	4	3.71 mJ/cm^2^	7.84 (−6 dB)	3.65	2.13 (peak)
[63]	MWCNT-integrated/PDMS MWCNT-xylene/PDMS (<1μm) MWCNT-gel/PDMS	SPOT-10-500-1064, Elforlight	1064	-	2	33.1 mJ/cm^2^	26.2 (−6 dB) 39.8 (−6 dB) 29.2 (−6 dB)	-	13.8 (peak) 12.2 (peak) 21.5
[13]	CNTs–PDMS (16 μm)	Surelite I-20, Continuum	-	20	6	42.4 mJ/cm^2^	> 15	-	> 50 (peak)
[1]	MWCNTs–PDMS	SPOT-10-500-1064, Elforlight	1064	8000	2	40 μJ	31.3 (−6 dB)	1.5	1.87
[60]	Functionalized CNTs, xylene, and PDMS	SPOT-10-500-1064, Elforlight	1064	100	2	96.1 mJ/cm^2^	20	-	4
[83]	CNTs–PDMS (10 μm)	SPOT-10-500-1064, Elforlight	1064	1000	2	41.6 mJ/cm^2^ 36.3 mJ/cm^2^	12 (−6 dB) 15 (−6 dB)	- -	3.6 (peak) 4.5 (peak)
[61,124]	CNTs–PDMS	Surelite, Inc	-	-	6	9.6 mJ/cm^2^	25 (−6 dB)	9.2	70 (peak)
[126]	MWCNTs (13.7 μm)–PDMS	SPOT-10-500-1064, Elforlight	1064	100	2	35 mJ/cm^2^	29 (−6 dB)	1.5	1.59 (peak-to-peak)
[131]	CNTs–PDMS	-	532	-	6	14, 16, 17.5, 18.5 mJ	-	<1	>30
[59]	MWCNT–PDMS	SPOT-10-500-1064, Elforlight	1064	50	2	20 μJ	26.5 (−6 dB)	1.5	8.8 (peak-to-peak)
[65]	MWCNTs–PDMS (49 μm)	FQS-400-1-Y-1064, Elforlight	1064	2000	<5	76 μJ	~27.1	2.7	~0.977
[158]	MWCNTs–PDMS (2–4 μm)	1047, Mosquito Innolas	1047	1000	11.4	12.7 mJ/cm^2^	1.5–12.7 MHz	4 mm	0.39–0.54 MPa
[133]	Photostable dye spliced device (<20 μm)–PDMS	SPOT-10-500-1064, Elforlight	1064	100	2	20.1 μJ	31.7 (−6 dB)	1.5	2.69 (peak-to-peak)

**Table 2 sensors-22-09541-t002:** PVDF membrane hydrophones employed in photoacoustic systems.

Refs	Model, Manufacturer	Detection Range [MHz]	Sensitivity [V/Pa]	Laser Fluence/ Energy	Aperture size [μm]	Distance [mm]	Acoustic Pressure [MPa]
[120,169]	HGL-0085, Onda	0.25–40	13	3.57 mJ/cm^2^	200	4.2	4.8
[122,169]	HGL-0085, Onda	0.25–40	13	1 mJ/cm^2^	-	7.5	3.78
[123,169]	HGL-0085, Onda	0.25–40	13	3.71 mJ/cm^2^	12 000	3.65	12.15 (peak)
[62,169]	HGL-0200, Onda	0.25–40	50	8.75 mJ/cm^2^	200	1	0.64 (peak-to-peak)
[116,169]	HGL-0200, Onda	0.25–40	50	13 mJ/cm^2^	-	1.8	0.18949
[119,170]	HMB-0500, Onda	0.5–45	631	56 mJ/cm^2^	-	2.85	~9 (peak)
[83,171]	75 μm, Precision Acoustics	1–30	~10	41.6 mJ/cm^2^ 36.3 mJ/cm^2^	-	0	3.6 (peak) 4.5 (peak)
[126,171]	75 μm, Precision Acoustics	1–30	~10	35 mJ/cm^2^	-	1.5	1.59 (peak-to-peak)
[133,172]	200 μm needle hydrophone, Precision Acoustics	0.1–40	55	20.1 μJ	-	1.5	2.69 (peak-to-peak)
[118,172]	200 μm needle hydrophone, Precision Acoustics	0.1–40	55	15 mJ/cm^2^	600	1.6	1.7 (peak)

**Table 3 sensors-22-09541-t003:** Resulting performance of photoacoustic-based ultrasound probes using Fabry-Pérot as ultrasound detectors.

Refs.	Material	Laser Model	Excitation Wavelength [nm]	Detector Bandwidth [MHz]	Laser Fluence/Energy/Power	Distance [mm]	Acoustic Pressure [MPa]
[128]	Polymer etalon structure with thickness of 5.9 μm	MIRVISION, Keopsys, Lannion	1511.5	~30–70	0.03 J/cm^2^	-	-
[63,191]	38 μm thick Parylene C polymer film spacer between two dichroic dielectric mirrors	Tunics T100S-HP CL, Yenista Optics	1500–1550	-	33.1 mJ/cm^2^	At the coating	12.2–21.5
[160]	Polymer spacer between two dielectric mirrors	LOTIS TII LS-2145-LT150	700–900 1064	23 (−3 dB)	<4 mJ/cm^2^ 20 mJ/cm^2^	-	-
[1]	Parylene C between two dielectric mirrors	Tunics T100S-HP CL, Yenista Optics	1500–1600	-	40 μJ	1.5	1.87
[59,192]	5 µm thick layer of Parylene C between two mirrors	Tunics T100S-HP CL, Yenista Optics	1520–1570	80	9 mW	1.5	8.8 (peak-to-peak)

## Data Availability

Not applicable.

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
