# Peer review of "A Comprehensive Review on Photoacoustic-Based Devices for Biomedical Applications"

_sensors, 2022, doi:10.3390/s22239541_

Round 1

Reviewer 1 Report

In the article, the authors perform an extensive review of the photoacoustic effect applications and devices which can be a very good starting for readers non-specialists in the field of photoacoustic technology.
I have only a few minimal comments:
-- Line 159.  Please, extend the phrase "target materials with certain properties", it is not unclear what are these properties.
-- Line 398.  " (Figure 10Figure 10)."
-- Please, clear the background of Figure 10.
-- PDMS term should be defined before subsection 3.2.2.

Reviewer 2 Report

In the Introduction: It is advisable to write down the structure of the Review: what topics or sections will be written down throughout the Paper.  

The Introduction also lacks background on the importance of the review of photoacoustic-based devices for biomedical applications 

Some sentences, such as Line 35-38; 209-215; 345-349 (check others): the sentence is too long, making it miss the topic. consider dividing it into 2-3 sentences. 

Line 124-127: electromagnetic radiation or light? both are different. be consistent. And change the sentence structure.

Some paragraphs only contained 1 sentence, such as 236-239; 266-268; 317-319 (check others): paragraph should contain more than 1 sentence.

Figures should be explained in sentences. Some figures are stand-alone without explanation. See Fig 4, Fig 5 (check others). Explain and discuss the figures related to the topic sentence or paragraph. 

343-345: check the sentence structure (the use of 'and')

Reviewer 3 Report

1、  Both photoacoustic and photoelectric devices are advanced technology for imaging applications (e.g. NATURE COMMUNICATIONS, 2021, 12, 6696). I suggest the authors to add some comments in the introduction portion to conpare the above two types of devices and at least refer the literatures of photoelectric devices to make this paper objective and complete.

2、  One key advantage of the endogenous photoacoustic agents is that they are naturally inside the body. However, as shown in Fig. 4, water has wide spectral absorption especially in the NIR region. So how to prevent the interference of water which is naturally inside the body and detect the useful information?

3、  Several figures are incomplete and not clear, e. g. Fig. 6.

4、  In the schematic of the experimental setup for an all-optical ultrasound probe (Fig. 20), a photodiode is used in this detection system. Since photoacoustic rather than photoelectric information is the one to be detected, what is the role for a photoelectric device?
